# SYNTHTOOLS: A FRAMEWORK FOR SCALING SYNTHETIC TOOLS FOR AGENT DEVELOPMENT

## ABSTRACT

AI agents increasingly rely on external tools to solve complex, long-horizon tasks. Effective development of such agents requires large-scale training in environments where they can safely practice using diverse tools, adapt strategies, and iteratively improve. However, real-world APIs are limited in availability, domain coverage, and stability, often requiring access keys and imposing rate limits, rendering them impractical for scalable training. To address these challenges, we introduce **SynthTools**, a flexible and scalable framework for generating synthetic tool ecosystems. Our framework consists of three core components: *Tool Generation* for automatic and scalable creation of diverse tools across domains, *Tool Simulation* to emulate realistic tool behaviors, and *Tool Audit* to ensure consistency and reliability. Using SynthTools, we generate large corpora of synthetic tools and tasks, enabling controllable, stable, and domain-agnostic training environments for LLM agents. By decoupling training from real-world API constraints, SynthTools provides stable interfaces, supports multi-domain experimentation, and thereby accelerates the development of robust, general-purpose LLM agents.

## 1 INTRODUCTION

LLM agents have garnered significant attention for their potential to tackle complex, real-world tasks. These modern agents are increasingly envisioned to leverage multiple tools in combination to solve complex, long-horizon problems (Xu et al., 2023; Qin et al., 2023; Yao et al., 2024). The proficiency in using tools has become a central capability for a performant agent. As evident throughout major advances in machine learning (Deng et al., 2009; Hoffmann et al., 2022), the scale and diversity of training are among the most critical factors influencing the quality of a model. To realize the vision of autonomous agents that can navigate sophisticated tool ecosystems and deploy resources effectively, it is essential to have access to comprehensive and diverse toolsets for training and experimentation.

However, agent development currently faces a significant bottleneck: in practice, tools are primarily accessed via APIs, yet real-world APIs remain scarce in both number and domain coverage. Much of the existing research focuses on curating these APIs or building high-fidelity replicas, but these collections still suffer from limited scope and diversity (Xu et al., 2023; Tang et al., 2023). For instance, ACEbench (Chen et al., 2025b) covers only eight broad domains, whereas $\tau$-bench (Yao et al., 2024) considers only two application domains.

Moreover, while real-world APIs offer authenticity, they come with practical constraints such as the need for API keys, usage limits, or rate throttling. Furthermore, actively maintained APIs are subject to frequent interface changes or deprecations, which can disrupt experimental configurations, destabilize training pipelines, and compromise reproducibility (Guo et al., 2025). These limitations render real APIs ill-suited for large-scale experimentation. This leads us to a fundamental question: how can we build a scalable and diverse tool ecosystems that support comprehensive agent training and evaluation?

**Synthetic tools** offer a flexible and controllable solution. Just as humans can learn from stylistic examples and generalize the acquired skills to real-world contexts, we conjecture that modern agents can similarly acquire sophisticated tool-use abilities by training on a diverse pool of synthetic tools and transferring learned capability to real-world interfaces. Recent studies provide supporting

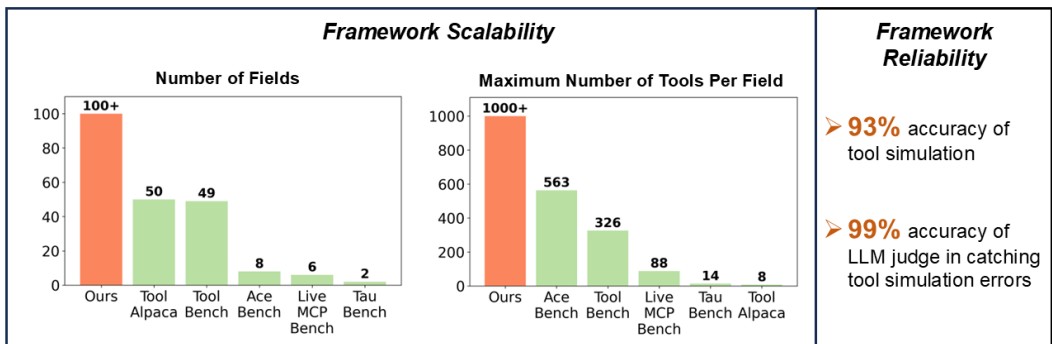

Figure 1: Scalability and reliability of our framework. Left: Our framework achieves noticeably higher scalability than existing benchmarks, both in the number of fields and the maximum tools supported per field. *Right* Both our tools generated with our framework, as well as the LLM judge built for tool audit, are highly reliable.

evidence: agents trained on synthetically generated tools or their outputs have demonstrated genuine learning of transferable tool-use skills, achieving reasonable performance on held-out tool-use dataset (Li et al., 2023; Kimi, 2025; Sullivan et al., 2025). Despite these promising developments, the community still lacks a clear and scalable way to create non-trivial synthetic tools for agent development. To lay the foundation for agents to fully realize the potential of this paradigm, our work aims to address this bottleneck by proposing a framework for reliable generation of complex synthetic tools at scale.

We propose **SynthTools**: a synthetic tool generation framework consisting of the following three components:

• **Tool Generation**: Automatically creating a wide variety of tools with diverse interfaces and functionalities (See Figure 2).

• **Tool Simulation**: Accurately emulating tool behaviors to closely mirror real-world interactions.

• **Tool Audit**: Careful and scalable assessment of the reliability and consistency of generated tools to ensure quality (See Figure 4).

Using this framework, we generate a large corpus of reliable synthetic tools spanning a wide range of domains. Building on these tools, we demonstrate how to construct realistic tasks for agent training and evaluation. Rather than offering a fixed set of tasks, our framework serves as a flexible foundation for others to create their own tools and tasks tailored to specific agentic capabilities they are interested in. Overall, SynthTools is scalable, flexible, and stable. By making scalable tool generation reliable and efficient, SynthTools help accelerate the development of robust, general-purpose LLM agents

## 1.1 RELATED WORK

**Agent Tool Use Evaluation.** Recent benchmarks such as APIBench, API-Bank, ToolBench, Tool-Alpaca, and ToolQA evaluate LLM agents on their tool-use capabilities. These benchmarks involve both real-world APIs (Li et al., 2023; Xu et al., 2023) and carefully curated synthetic API simulators tailored to specific domains (Patil et al., 2024; Tang et al., 2023; Zhuang et al., 2023; Wang et al., 2023; Yao et al., 2024; Chen et al., 2025b; Guo et al., 2025). With the emergence of MCP servers, some recent studies have shifted focus toward evaluating agents using tools hosted on MCP servers (Mo et al., 2025; Yin et al., 2025). While these efforts provide valuable benchmarks for agent performance, they often face scalability challenges due to the limited number and diversity of tools. Moreover, both real APIs and MCP servers present practical obstacles such as access restrictions, rate limitations, and unstable interfaces.

**Fine-Tuning Agents for Tool Use.** Fine-tuning LLMs on curated tool-use datasets has shown promising results. For instance, ToolLLAMA constructs a large-scale dataset from real tools avail-

able on RapidAPI Hub and fine-tunes models to perform tool-augmented tasks (Qin et al., 2023). Similarly, Fang et al. (2025) focus on scaling the number of tasks by aggregating real-world APIs from sources such as (Qin et al., 2023; Prabhakar et al., 2025), thereby enhancing the breadth of fine-tuning. Other studies (Kimi, 2025; Sullivan et al., 2025) explore synthetic tool environments for fine-tuning, demonstrating the potential of such tools. However, the dataset introduced by Kimi (2025) is proprietary, and the focus of Sullivan et al. (2025) is on deeply compositional toolchains rather than diversity, thus limiting its generalizability. Building on these insights, we introduce an open-source framework for generating synthetic tools that are both diverse and deeply composable, enabling scalable training and evaluation of LLM agents in tool-use settings.

| | Generated / Curated | Field Scalability | Tool Scalability | Complex Tools | Interactive |
|---|---|---|---|---|---|
| **SynthTools (Ours)** | Generated | ✓ | ✓ | ✓ | ✓ |
| $\tau$-Bench | Curated | ✗ | ✗ | ✗ | ✓ |
| StableToolBench | Curated | ✗ | ✗ | ✗ | ✓ |
| AceBench | Generated | ✗ | ✗ | ✗ | ✗ |
| ToolAlpaca | Generated | ✓ | ✓ | ✗ | ✓ |
| RandomWorld | Generated | ✗ | ✓ | ✗ | ✓ |

Table 1: Our framework is designed with inherent scalability and interactivity. By iteratively refining our pipeline, we enable our framework to generate highly complex tools.

## 2 TOOL GENERATION

Agentic systems rely heavily on the breadth and scale of the tools they can access. However, generating diverse tools at scale remains a major challenge. Manually engineering a diverse and extensible toolset to support large-scale training of agentic LLMs is impractical. A promising alternative is to harness the generative capabilities of LLMs themselves to synthesize tools. Yet, naively generating tools via LLMs often results in redundant or aimless toolsets—tools that fail to target meaningful tasks and lack structural interconnections. This leads to weakly compositional tool chains that are only capable of solving trivial problems.

To address this limitation, we propose a scalable framework for synthesizing large, diverse, and non-redundant toolsets grounded in meaningful tasks and workflows. Our approach employs a hierarchical domain evolution procedure (Figure 2) that systematically refines a broad domain into a concrete and coherent toolset. This process is anchored in practitioner workflows as simulated by LLMs. Starting from a general domain, we progressively decompose it into subdomains, then into task families, and finally into specific tools whose interfaces reflect domain-relevant constraints and interactions. Concretely, the framework proceeds as follows:

**Field → Sub-domain.** Given a seed set of fields (e.g., *healthcare*, *finance*, *materials science*), we prompt a large language model (LLM) to propose coherent subdomains that: (a) partition typical workflows; (b) surface stakeholders and entities operating in the field; and (c) admit meaningful, tool-addressable operations.

**Sub-domain → Task.** For each subdomain, the model proposes task families. Each task can be thought of as a , yielding natural task statements suitable for documentation and test-case generation.

**Task → Tool.** Tasks are then realized as concrete tools. Tools are encouraged to be *composable*: each tool advertises upstream dependencies (what it consumes) and downstream affordances (what it produces), enabling multi-tool plans. The interface includes name, description, parameters, failure model, and output details.

We use targeted prompting at each stage to control diversity, complexity, and input/output (I/O) characteristics of the tools. Our *tool* abstraction is characterized by: (i) a name and natural-language description; (ii) a parameter schema; and (iii) an I/O contract (preconditions, postconditions, and error modes). Formally, a tool is a tuple (`name`, `description`, `parameters`, `usage`,

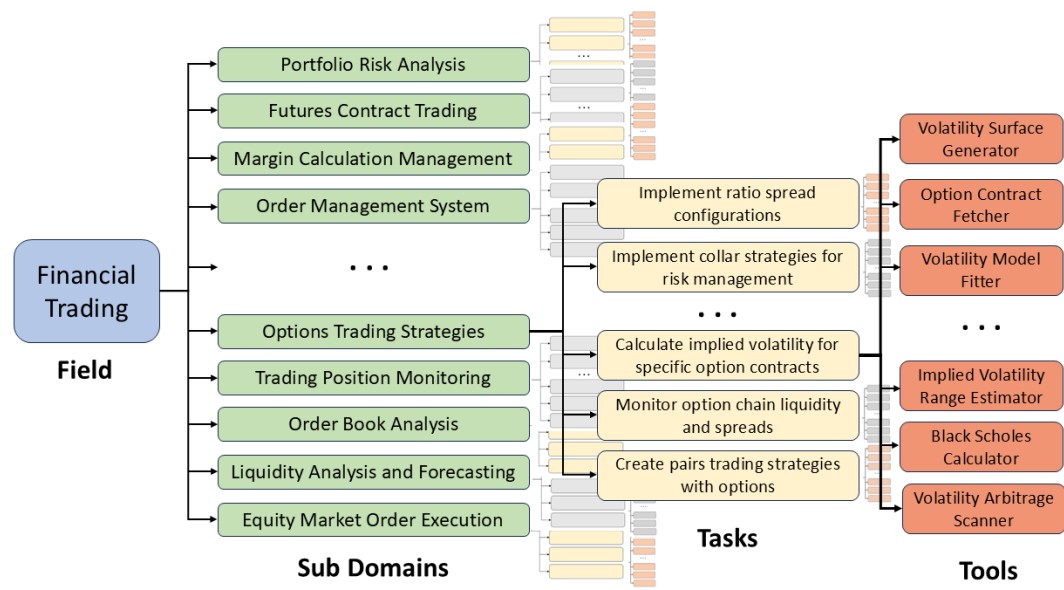

Figure 2: An example of tool generation through hierarchical domain evolution procedure.

failure_modes, output_schema). Figure 3 illustrates a tool generated by our pipeline in the e-commerce and retail domain. Additional examples are provided in Section B.

As shown in Figure 1, our framework demonstrates strong scalability by generating a toolset that spans 100 distinct fields, with each field potentially comprising up to 1,000 tools.

**Deduplication.** Since our tools are generated through a hierarchical process, there is a possibility of creating duplicate entries due to overlapping workflows and functionalities across sub-domains and tasks. To address this, we apply a deduplication procedure designed to identify and remove near-duplicate tools. This procedure leverages semantic similarity, computed using LLM embeddings of the tool descriptions. Deduplication is a crucial step to ensure the resulting dataset is of high quality and suitable for both training and evaluation purposes. Full details of the deduplication process are provided in Appendix D.

## 3 TOOL SIMULATION

Once the tools have been generated, the next step is to systematically emulate API call behavior for each tool configuration (Figure 3). A reliable tool simulator must return appropriate error messages for incorrect or incomplete calls and generate valid responses for correct ones. Achieving this behavior consistently in practice is challenging.

To address this, we decompose the simulation procedure into two distinct stages: **parameter validation** and **response generation**. In both stages, we prompt a large language model (LLM)—with access to the tool configuration—to emulate tool behavior. The two stages are as follows:

1. **Parameter validation.** In this stage, the simulator emulates an API gateway by ensuring that all schema and structural constraints are satisfied. It begins by verifying the tool name, then checks for the presence of all required parameters, correct data types, mutual consistency, and any cross-field constraints. If any condition fails, the simulator returns a specific error message identifying the first issue encountered, along with the corresponding HTTP status code mirroring conventional API gateway behavior.

2. **Response generation for valid calls.** If parameter validation is successful, the simulator advances to the response generation stage. Depending on the nature of the tool call, this stage proceeds via either **data generation** or **information deduction**. For tool calls that require generating new data based on the input parameters, the simulator produces realistic outputs that adhere

---

**Field to Tool Evolution**

**Field:** Ecommerce and Retail → **Sub Domain**: Product Catalog Management → **Task:** Product Information Creation and Updates → **Tool:** Product Data Validator

---

**The Generated Tool**

**Tool Name:** Product Data Validator
**Description:** Validates product information against predefined schemas and business rules to ensure data quality and completeness before catalog updates.

**Parameters:**
- **product_name** (string, required) – The name of the product to validate
- **product_attributes** (array of strings, required) – List of product attributes in `key:value` format
- **validation_level** (string, optional, default = `standard`) – Validation strictness: `basic`, `standard`, or `strict`

**Error Messages:**
- Missing required fields: Ensure `product_name` and `product_attributes` are provided.
- Invalid validation level: Use one of [`basic`, `standard`, `strict`].
- Malformed attributes: Attributes must be in `key:value` string format.
- Empty product data: Product name cannot be empty or null.

**Usage:** Provide `product_name` and `product_attributes` array, optionally set `validation_level`. Returns validation status and any detected issues.

**Output Details:**
- **validation_status** (string) – Overall validation result: `passed`, `failed`, or `warning`
- **issues_found** (array of strings) – List of validation issues or errors detected
- **completeness_score** (number) – Data completeness percentage (0–100)

Figure 3: An example of tool generated through our pipeline for the e-commerce and retail field.

to predefined schemas and domain-specific patterns. In contrast, when the tool call demands reasoning over metadata and the initial configuration, the simulator systematically cross-references these inputs to infer the current system state, relevant entities, and functional behavior. It then synthesizes this information to logically derive the precise response the API would produce under the given conditions.

**Refining tool simulator prompts**   To ensure reliable simulator behavior, we refined its prompts through extensive manual testing. This was an inherently iterative process: we repeatedly updated the prompt, tested it manually, analyzed failure cases, and adjusted the prompt accordingly (See Section C.2 for the final version). We then evaluated the finalized prompts on a set of SynthTools generated using our framework. This evaluation was conducted manually. The final refined prompt achieved an accuracy of **93.6%**, as verified across 200 tool responses.

To further assess the simulator's performance, we evaluated it against ACEBench (Chen et al., 2025a)—a suite of sandbox tools with deterministic, programmatically defined behaviors. We aligned our simulator's configuration to ACEBench specifications, ensuring consistent initial states. We then compared the simulator's responses to ACEBench's ground truth outputs. In total, we generated approximately eight test calls per tool across 20 ACEBench tools (161 calls overall), encompassing both successful executions and various failure scenarios. The simulator matched the ground truth in 151 out of 161 cases, yielding a 94% accuracy. 14 (out of 20) tools showed perfect agreement across all test cases. The 10 mismatches were mostly due to differences in implementation-specific prioritization (e.g., whether authentication is checked before or after parameter validation) rather than fundamental flaws in simulation logic. See Figure 5 for illustrative examples.

We further assess the simulator at scale using an LLM-based evaluator, as described in the following section

| Tool Config Source | **Performance** | **Evaluation Method** |
| --- | --- | --- |
| SynthTools | 93.6% | Manual verification |
| ACEbench | 94% | Ground-truth comparison |

Table 2: Performance of tool simulator

.

## 4 TOOL AUDIT

To ensure high-quality tools, we subject each generated tool to a rigorous quality control pipeline consisting of systematic test case construction and LLM-based validation. We detail this process below.

**Test Case Generation.** We test each generated tool with a comprehensive suite of test calls spanning four distinct modes. Tools that fail more than 1 tests are discarded. Specifically, we evaluate each tool under the following scenarios:

1. **Schema failures (basic parameter validation).** The tool call cannot be parsed or validated at the programmatic level due to missing required parameters, incorrect parameter types, or malformed inputs that prevent basic function invocation.

2. **Constraint failures (tool-specific validation errors).** The parameter schema is satisfied, but tool-specific constraints are violated, such as mismatched array lengths, invalid value ranges, or logical inconsistencies between parameters that the tool specification explicitly prohibits.

3. **Execution failures (metadata constraint violations).** The tool call passes all parameter validation but contradicts the current system state or metadata during execution, such as referencing non-existent user records, attempting operations on unavailable resources.

4. **Successful executions.** The tool call is valid at all levels and the simulator should return responses that are consistent with the tool specification and provided metadata, correctly performing operations or returning requested data.

---

**Failure mode 2: Parameter inconsistency**

**Tool call message:**
*Insurance_Information_Updater* (patient_id = 'PAT001', insurance_fields = ['provider', 'policy_number'], insurance_values = ['Blue Cross'])

**Response:**
*Status*: FAIL, *Status Code*: 400, *Error Message*: Mismatched fields and values: Ensure insurance_fields and insurance_values arrays have the same length.

---

**Success mode: Correct response**

**Tool call message:**
*Insurance_Information_Updater* (patient_id = 'PAT001', insurance_fields = ['provider'], insurance_values = ['Blue Cross'])

**Response:**
*Status*: PASS, *Status Code*: 200, *Return Data*: update_status: Success, updated_insurance: ['provider']

---

In the boxes above, we present representative examples demonstrating how the simulator handles a variety of testing scenarios. These include both successful executions and a failure case along with its corresponding error response (see Appendix C.1 for additional examples). For each tool, we prompt a large language model (LLM) to generate 2–3 test calls per failure mode. These calls are then executed through our tool simulator, and the resulting responses are recorded.

**LLM-Based Verification.** To assess response correctness, we employ a carefully engineered *LLM judge*. This judge receives the tool specification, test call, and simulator response as input, then

returns a structured judgment comprising correct/incorrect status, confidence score, and detailed rationale for correctness and compliance assessment.

We iteratively refined the judge through testing with hand-crafted edge cases (e.g., subtle type violations, cross-field incompatibilities) until it reliably identified each error type with appropriate rationale. We conducted stress testing of the judge across six distinctive failure and success modes to ensure judge reliability. We explain the procedure and results in detail in Section 5.

Among the 3,300 tool responses, we found that 3,054 were correct, corresponding to an accuracy of 93% (as evaluated by the LLM judge). These responses came from 352 tools, of which 89% made no more than one error across 8–10 stress test calls (see Table 3). Regarding the errors, 56% were due to Failure Mode 2, 34% to Failure Mode 3, and 10% to Failure Mode 1.

| Number of Incorrect Responses | 0 | 1 | 2 | 3 |
|---|---|---|---|---|
| Percentage of Tools | 57% | 32% | 7% | 3% |

Table 3: Distribution of incorrect responses among 8-10 stress test calls across 352 tools

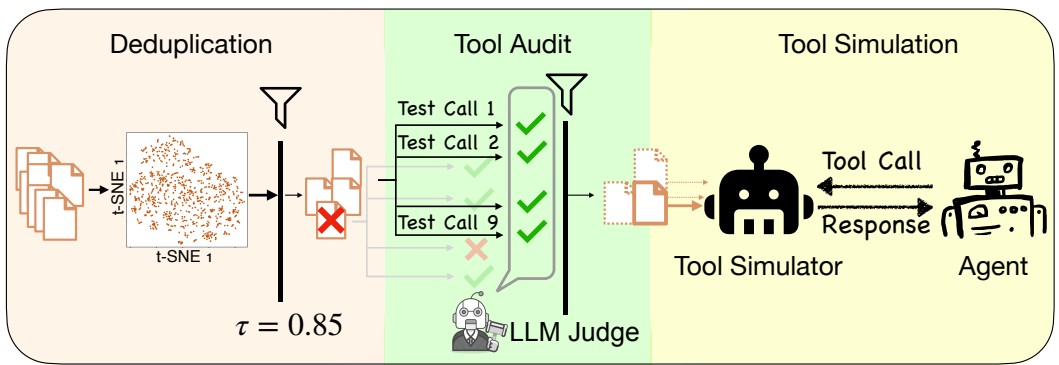

Figure 4: Our Tool Audit component ensures the quality of the tools after deduplication.

## 5 HOW RELIABLE IS THE LLM JUDGE?

The integrity of our framework critically depends on the reliability of the LLM judge. In this section, we evaluate how effective the judge is in distinguishing correct and incorrect tool simulator responses. To this end, we construct six stress-test scenarios, organized around three categories of tool call failures—schema, constraint, and execution. For each category, we include both valid and invalid simulator outputs. Through this stress-testing process, we aim to confirm that the judge can consistently identify when the simulator is functioning as intended versus when it produces erroneous outputs.

We manually verified 300 stress test cases. Among these, the judge made only 3 errors, yielding an accuracy rate of 99%. This demonstrates that our quality control process can reliably distinguish between well-functioning and problematic tool simulations, making it suitable for large-scale filtering. Importantly, since our framework generates tools at scale, minimizing false positives is critical; passing problematic tool responses through quality control would undermine reliability. Notably, we observed a false positive rate of 0%—the judge successfully identified all incorrect simulator behaviors. Together, these results establish the robustness of our judge in diagnosing tool simulator performance and reinforce confidence in its role within our tool audit pipeline.

| Metric | Accuracy | False Positive Rate |
|---|---|---|
| Performance | 99% | 0% |

Table 4: Performance of LLM judge under stress tests

**ACEBench test example**

**Tool call message:**
*modify_flight*(user_id = 'user1', reservation_id = 'res_1', new_cabin = 'Business Class')

**Response:**
*Status*: PASS, *Status Code*: 200, *Return Data*: Cabin upgraded to Business Class. Price difference of 1800 yuan has been charged. Modification completed successfully.

**Sandbox execution message:** Cabin change successful. Price difference paid: 1800.

**Judge success example**

**Tool call message:**
*Performance_Metrics_Calculator*(start_date = '2024-01-31T23:59:59Z', end_date = '2024-01-01T00:00:00Z', **...**)

**Response:**
*Status*: PASS, *Return Data*: ...

**Judge Reasoning:**
The tool call has incorrectly specified start_date='2024-01-31T23:59:59Z' being later than the end_date='2024-01-01T00:00:00Z'. The simulator should have returned FAIL with this error message.

Figure 5: *Left*: An example ACEBench tool call with simulator return data precisely matching the execution output. *Right:* An example where the judge correctly identifies an erroneous response.

## 6    TOOL SET AND TASKS

Using our pipeline, we generate a large corpus of synthetic tools spanning multiple domains, including e-commerce and retail, healthcare, financial trading, and more. We demonstrate the scalability of our pipeline along two key dimensions:

- **Scaling through diversification of fields:** To demonstrate the flexibility of our pipeline in generating tools for diverse domains, we create 50 tools each across 100 different fields. Figure 6 illustrates the breadth and diversity of the resulting toolset for 37 fields. See Appendix E for the full 100-field figure. While the selected domains are not exhaustive, this experiment shows that our pipeline can effectively scale the generation of tools across a wide variety of application areas.

- **Scaling the number of tools within a field:** We also investigate the ability of our pipeline to scale tool generation within a specific field. A key question in this setting is whether increasing the number of tools results in genuinely novel tools or merely duplicates. To explore this, we focus on the e-commerce and retail domain and scale the tool count up to 1,000. As shown in Figure 7, the distribution indicates a high degree of tool uniqueness. We further validate this trend by generating 200 tools in several other domains, observing consistent results. This supports the conclusion that our pipeline can effectively scale the number of tools within a single field without significant redundancy.

During the deduplication stage, we filtered out approximately 9% of near-duplicate tools. In a subsequent tool auditing phase (Section 4), we discarded an additional $\approx 11\%$ of tools from a sampled subset, based on failures in consistency checks or violations of interface contracts. These steps result in a carefully curated collection of high-quality tools, suitable for downstream training and evaluation tasks.

### 6.1    TASKS

We demonstrate how to construct multi-step and multi-turn tasks that require the use of multiple tools and structured decision-making by an agent using our pipeline. Our hierarchical domain-evolution procedure naturally produces tasks with intermediate subgoals requiring deep composable tools (Section A); we leverage these to derive specific tasks that can be used for training/evaluation environments. A critical component is the generation of meta-data consumed by the tool simulator to produce grounded responses on valid calls. This meta-data is generated in parallel with task creation to ensure consistency between task requirements and tool outputs.

We provide illustrative examples of (i) the associated meta-data, (ii) task specifications, and (iii) the minimal tool set required to solve each task (Section A). For experimental settings, one can train/evaluate agents under two regimes: (a) an *exact* tool set containing only the tools needed for

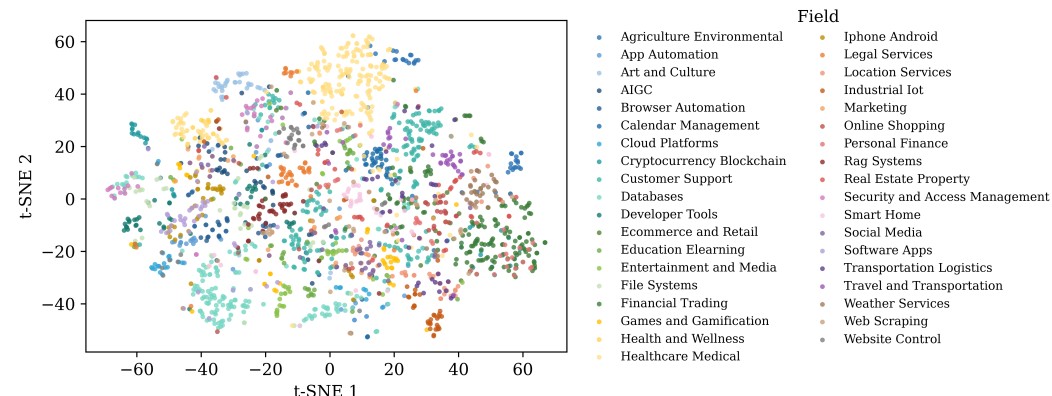

Figure 6: Distribution of tool embeddings across diverse field.

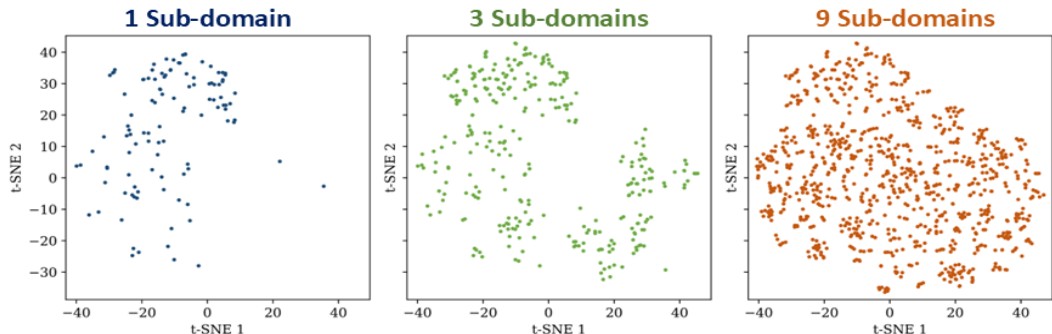

Figure 7: **Scaling the number of tools within a field (e-commerce and retail):** As we scale the number of tools within a field, they get more diverse rather than just producing duplicates. (left: 1 sub-domain, 110 tools, center: 3 sub-domains, 315 tools, right: 1 sub-domain, 933 tools)

the task, and (b) an *extended* tool set that includes distractor tools. The latter probes the agent's ability to discover and select the appropriate tools under realistic ambiguity.

As a complementary approach, one can also explore tasks generated directly by LLMs conditioned on a given tool set. While this approach can increase variety, we find it less reliable than the hierarchical construction: LLM-generated tasks may under-specify dependencies or omit long-horizon structure, whereas the hierarchical procedure yields more coherent, multi-tool plans.

# 7 CONCLUSION

We introduce SynthTools, a scalable pipeline for generating synthetic tool ecosystems to support the development of tool-using LLM agents. Our approach integrates hierarchical tool generation, simulation of realistic behaviors, and rigorous quality control to produce diverse, reliable, and reusable toolsets. Experiments demonstrate that the pipeline scales across domains, maintains high simulation fidelity, and enables the construction of complex, multi-step tasks for training and evaluation. By decoupling agent training from the limitations of real-world APIs, SynthTools provides a stable and flexible foundation for advancing research in general-purpose agentic systems. We hope this framework will catalyze broader exploration of synthetic environments and foster progress toward robust, adaptive LLM agents.

## 8 REPRODUCIBILITY STATEMENT

We provide our code in this anonymized repository: https://anonymous.4open.science/r/SynthTools-44C6/README.md

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

## A   EXAMPLE TASKS

### A.1   EXAMPLE 1 (FIELD: ECOMMERCE AND RETAIL)

---

**Multi Tool Workflows**

**Field:** Ecommerce and Retail → **Subfield:** Product Catalog Management → **Task:** Product Information Creation and Updates

**List All Tool Names and Dependencies Gnereated under this task**

1. Product Data Validator – raw product data → validated product records
2. Product Content Generator – specs → descriptions, titles, keywords
3. Image Processor – raw images → optimized product images
4. Pricing Calculator – cost data, rules → calculated prices
5. Inventory Sync Checker – product IDs → stock status validation
6. Category Classifier – product attributes → category assignments
7. SKU Generator – product details → unique SKU codes
8. Bulk Import Processor – CSV/Excel files → processed product batches
9. Product Comparison Tool – multiple product records → difference reports
10. Catalog Publisher – finalized products → multi-channel updates
11. Product Search Optimizer – product data → SEO metadata
12. Quality Assurance Scanner – product records → quality issue reports

**Simple Multi-Tool Workflows:**
1. *Single product creation:* SKU Generator → Product Data Validator → Product Content Generator → Catalog Publisher
2. *Image optimization*: Image Processor → Product Data Validator → Catalog Publisher

**Medium Multi-Tool Workflows:**
1. *New product launch:* SKU Generator → Product Data Validator → Category Classifier → Product Content Generator → Image Processor → Pricing Calculator → Product Search Optimizer → Catalog Publisher
2. *Inventory synchronization:* Inventory Sync Checker → Product Data Validator → Product Comparison Tool → Catalog Publisher

**Complex Multi-Tool Workflows:**
1. *Bulk catalog update:* Bulk Import Processor → Product Data Validator → Category Classifier → Product Content Generator → Image Processor → Pricing Calculator → Inventory Sync Checker → Product Search Optimizer → Quality Assurance Scanner → Product Comparison Tool → Catalog Publisher
2. *Complete catalog audit:* Quality Assurance Scanner → Product Comparison Tool → Product Data Validator → Category Classifier → Pricing Calculator → Inventory Sync Checker → Product Search Optimizer → Catalog Publisher

---

**Meta Data**

**Products**:

**1. Product-id**: P1001, **product-name**: Wireless Noise-Cancelling Headphones, **brand**: AudioMax, **category**: Electronics, **attributes**: [color: Black, battery-life: 30h, wireless: true], **specifications**: [Bluetooth 5.0, ANC, USB-C charging], **base-cost**: 75, **markup-percentage**: 40, **currency**: USD, **competitor-prices**: [129.99, 139.99, 119.99], **stock**: [warehouse-A: 120, warehouse-B: 30]

**2. Product-id**: P1002, **product-name**: Ergonomic Office Chair, **brand**: ComfortPro, **category**: Furniture, **attributes**: [color: Grey, adjustable: true, material: Mesh], **specifications**: [Adjustable height, Lumbar support, 360 swivel], **base-cost**: 95, **markup-percentage**: 50, **currency**: USD, **competitor-prices**: [199.99, 189.99, 210.00], **stock**: [warehouse-A: 15, warehouse-B: 5]

**3. Product-id**: P1003, **product-name**: Smart LED Light Bulb, **brand**: BrightLite, **category**: Home Appliances, **attributes**: [color: RGB, connectivity: WiFi, power: 9W], **specifications**: [App control,

Voice assistant compatible, Dimmable], **base-cost**: 8, **markup-percentage**: 100, **currency**: USD, **competitor-prices**: [14.99, 12.99, 16.99], **stock**: [warehouse-A: 500, warehouse-B: 200]

**4. Product-id**: P1004, **product-name**: Sports Running Shoes, **brand**: FlexiRun, **category**: Footwear, **attributes**: [color: Blue, size: 10, material: Mesh], **specifications**: [Lightweight sole, Breathable, Shock absorption], **base-cost**: 40, **markup-percentage**: 60, **currency**: USD, **competitor-prices**: [79.99, 74.99, 69.99], **stock**: [warehouse-A: 200,warehouse-B: 150]

**5. Product-id**: P1005, **product-name**: 4K Ultra HD Smart TV, **brand**: VisionX, **category**: Electronics, **attributes**: [size: 55inch, resolution: 4K, smart: true], **specifications**: [HDR10+, HDMI 2.1, Dolby Atmos], **base-cost**: 300, **markup-percentage**: 35, **currency**: USD, **competitor-prices**: [449.99, 499.99, 479.99], **stock**: [warehouse-A: 25, warehouse-B: 10]

**Existing categories**: Electronics, Furniture, Home Appliances, Footwear, Sports Equipment, Toys, Accessories

**Warehouses**: warehouse-A, warehouse-B, warehouse-C

**Channels**: website, amazon, ebay, mobile-app

**Import files**:
**1. File-path**: /data/batch1-products.csv, **file-format**: csv, **column-mapping**: [name: product-name, brand: brand, category: category, price: base-cost, stock: inventory]

**2. File-path**: /data/batch2-products.json, **file-format**: json, **column-mapping**: [title: product-name, company: brand, type: category, cost: base-cost, inventory: stock]

## Tasks

**Task 1**: Create a new SKU and publish the Wireless Noise-Cancelling Headphones to website and Amazon with validated product data and generated content.
**Info**: Task-difficulty: simple, Task-validity: valid, Number-tools-required: 4
**Tools-required**: [SKU Generator, Product Data Validator, Product Content Generator, Catalog Publisher]

**Task 2**: Optimize and publish the Ergonomic Office Chair images to the website after validation.
**Info**: Task-difficulty: simple, Task-validity: valid, Number-tools-required: 3
**Tools-required**: [Image Processor, Product Data Validator, Catalog Publisher]

**Task 3**: Launch Smart LED Light Bulb with SKU creation, validation, category assignment, content, image optimization, pricing, SEO, and publish to all channels.
**Info**: Task-difficulty: medium, Task-validity: valid, Number-tools-required: 8
**Tools-required**: [SKU Generator, Product Data Validator, Category Classifier, Product Content Generator, Image Processor, Pricing Calculator, Product Search Optimizer, Catalog Publisher]

**Task 4**: Process the bulk import file batch1-products.csv, validate and classify products, generate content and optimized images, calculate pricing, check inventory, optimize for search, scan for quality issues, compare against existing catalog, and publish.
**Info**: Task-difficulty: complex, Task-validity: valid, Number-tools-required: 11
**Tools-required**: [Bulk Import Processor, Product Data Validator, Category Classifier, Product Content Generator, Image Processor, Pricing Calculator, Inventory Sync Checker, Product Search Optimizer, Quality Assurance Scanner, Product Comparison Tool, Catalog Publisher]

**Task 5**: Generate customer product reviews for the Sports Running Shoes and publish them on Amazon.
**Info**: Task-difficulty: simple, Task-validity: invalid, Number-tools-required: 2
**Tools-required**: [Review Generator, Catalog Publisher]

**Task 6**: Perform a complete catalog audit including legal compliance verification using a Legal Compliance Checker before publishing.
**Info**: Task-difficulty: complex, Task-validity: invalid, Number-tools-required: 9
**Tools-required**: [Quality Assurance Scanner, Product Comparison Tool, Product Data Validator, Category Classifier, Pricing Calculator, Inventory Sync Checker, Product Search Optimizer, Legal Compliance Checker, Catalog Publisher]

**Task 7**: Process an external XML product feed and update the catalog.
**Info**: Task-difficulty: complex, Task-validity: invalid, Number-tools-required: 2
**Tools-required**: [XML Feed Processor, Catalog Publisher]

## A.2 EXAMPLE 2 (FIELD: FINANCIAL TRADING)

### Multi Tool Workflows

**Field:** Financial Trading → **Subfield:** Trade Execution and Order Management → **Task:** Real-time Order Status Monitoring and Execution Tracking

**List All Tool Names and Dependencies Generated under this task**

1. Order Status Fetcher – order IDs, broker credentials → current order status data
2. Execution Event Logger – execution data, timestamps → logged execution records
3. Fill Notification Parser – raw broker messages → structured fill data
4. Order Latency Analyzer – order timestamps, execution data → latency metrics
5. Position Reconciler – order fills, current positions → reconciled position data
6. Alert Rule Engine – order status, thresholds → alert notifications
7. Multi-Broker Status Aggregator – multiple broker feeds → unified status view
8. Order Performance Calculator – execution data, benchmarks → performance metrics
9. Risk Exposure Monitor – open orders, positions → risk exposure data
10. Execution Quality Analyzer – fills, market data → execution quality scores
11. Order History Tracker – order events → complete order lifecycle data
12. Real-time Dashboard Generator – aggregated data → dashboard summaries

**Simple Multi-Tool Workflows:**
1. *Basic order tracking:* Order Status Fetcher → Execution Event Logger → Order History Tracker
2. *Fill monitoring:* Fill Notification Parser → Position Reconciler → Alert Rule Engine

**Medium Multi-Tool Workflows:**
1. *Performance monitoring:* Order Status Fetcher → Order Performance Calculator → Execution Quality Analyzer → Real-time Dashboard Generator
2. *Risk monitoring:* Multi-Broker Status Aggregator → Risk Exposure Monitor → Alert Rule Engine → Real-time Dashboard Generator

**Complex Multi-Tool Workflows:**
1. *Complete execution analysis:* Order Status Fetcher → Fill Notification Parser → Order Latency Analyzer → Order Performance Calculator → Execution Quality Analyzer → Position Reconciler → Real-time Dashboard Generator
2. *Multi-broker risk management:* Multi-Broker Status Aggregator → Order Status Fetcher → Risk Exposure Monitor → Position Reconciler → Alert Rule Engine → Execution Event Logger → Order History Tracker

### Meta Data

**Tool Sequences**:

Easy: [Order Status Fetcher → Execution Event Logger → Order History Tracker, Fill Notification Parser → Position Reconciler → Alert Rule Engine]

Medium: [Order Status Fetcher → Order Performance Calculator → Execution Quality Analyzer → Real-time Dashboard Generator, Multi-Broker Status Aggregator → Risk Exposure Monitor → Alert Rule Engine → Real-time Dashboard Generator]

Complex: [Order Status Fetcher → Fill Notification Parser → Order Latency Analyzer → Order Performance Calculator → Execution Quality Analyzer → Position Reconciler → Real-time Dashboard Generator, Multi-Broker Status Aggregator → Order Status Fetcher → Risk Exposure Monitor → Position Reconciler → Alert Rule Engine → Execution Event Logger → Order History Tracker]

**Trading Accounts**:

ACC_001_INST – Interactive Brokers, Institutional, Portfolio: 50,000,000, Risk: [max_pos: 5,000,000, sector: 15%, daily_loss: 500,000]

ACC_002_HEDGE – Alpaca, Hedge Fund, Portfolio: 25,000,000, Risk: [max_pos: 2,500,000, sector: 20%, daily_loss: 300,000]

ACC_003_PROP – Binance, Proprietary, Portfolio: 10,000,000, Risk: [max_pos: 1,000,000, sector: 25%, daily_loss: 150,000]

**Orders**:

ORD_001 – AAPL Buy 10,000 @ 175.50 (limit), Status: partially_filled (6,500 filled, 3,500 remaining), Avg Fill: 175.48, Venue: NASDAQ, Fees: 45.50

ORD_002 – TSLA Sell 5,000 (market), Status: filled, Avg Fill: 248.75, Venue: NYSE, Fees: 62.19

ORD_003 – MSFT Buy 8,000 (stop-limit 420/421), Status: pending, Venue: NASDAQ

ORD_004 – GOOGL Buy 2,500 @ 142.50 (limit), Status: rejected (reason: insufficient buying power), Venue: NASDAQ

ORD_005 – NVDA Sell 3,000 @ 875.00 (limit), Status: cancelled (1,200 filled @ 874.95), Venue: NASDAQ, Fees: 31.50

**Market Data (15:30 UTC)**:

AAPL – 175.52 (Bid 175.50, Ask 175.53, VWAP 175.45, TWAP 175.48, Vol 45M)

TSLA – 248.80 (Bid 248.75, Ask 248.85, VWAP 248.70, TWAP 248.72, Vol 32M)

MSFT – 419.75 (Bid 419.70, Ask 419.80, VWAP 419.80, TWAP 419.85, Vol 28M)

**Positions**:

ACC_001_INST – AAPL: 156,500 @ 172.30, Unrealized PnL: 503,680

ACC_002_HEDGE – TSLA: -8,200 @ 252.10, Unrealized PnL: 27,470

**Alert Rules**:

High Latency – ack_latency_ms ¿ 500 (priority: high)

Large Slippage – slippage_bps ¿ 50 (priority: critical)

Risk Limit Breach – exposure_percentage ¿ 80 (priority: critical)

**Execution Venues**:

NASDAQ – Exchange, Latency 12ms, Fill Rate 0.95, Impact 1.2

NYSE – Exchange, Latency 15ms, Fill Rate 0.92, Impact 1.1

Dark Pool 1 – Dark Pool, Latency 25ms, Fill Rate 0.78, Impact 0.8

**Broker Configurations**:

Interactive Brokers – api.ib.com, Orders/sec: 50, Types: [market, limit, stop, stop_limit], Status: active

Alpaca – paper-api.alpaca.markets, Orders/sec: 200, Types: [market, limit, stop], Status: active

Binance – api.binance.us, Orders/sec: 100, Types: [market, limit, stop_limit], Status: maintenance

## Tasks

**Task 1**: Track status of ORD_001 (IBKR), log execution events, update order history.
**Info**: Difficulty: easy, Validity: valid, Tools: 3
**Tools-required**: [Order Status Fetcher, Execution Event Logger, Order History Tracker]

**Task 2**: Parse TSLA ORD_002 fill, reconcile position, generate alerts.
**Info**: Difficulty: easy, Validity: valid, Tools: 3
**Tools-required**: [Fill Notification Parser, Position Reconciler, Alert Rule Engine]

**Task 3**: Retrieve ORD_001 status + auto hedge trade (invalid).
**Info**: Difficulty: easy, Validity: invalid, Tools: 4
**Tools-required**: [Order Status Fetcher, Execution Event Logger, Order History Tracker, Auto Trade Executor]

**Task 4**: Parse fills + SMS alerts (invalid).
**Info**: Difficulty: easy, Validity: invalid, Tools: 3
**Tools-required**: [Fill Notification Parser, Position Reconciler, SMS Alert Sender]

**Task 5**: Execution performance analysis for ORD_002 vs VWAP.
**Info**: Difficulty: medium, Validity: valid, Tools: 4
**Tools-required**: [Order Status Fetcher, Order Performance Calculator, Execution Quality Analyzer, Real-time Dashboard Generator]

**Task 6**: Aggregate status from brokers, monitor ACC_001_INST risk, alert + dashboard.
**Info**: Difficulty: medium, Validity: valid, Tools: 4
**Tools-required**: [Multi-Broker Status Aggregator, Risk Exposure Monitor, Alert Rule Engine, Real-time Dashboard Generator]

**Task 7**: Aggregate broker status + auto portfolio rebalance (invalid).
**Info**: Difficulty: medium, Validity: invalid, Tools: 4
**Tools-required**: [Multi-Broker Status Aggregator, Risk Exposure Monitor, Alert Rule Engine, Portfolio Rebalancer]

**Task 8**: Fetch performance + compliance reporting (invalid).
**Info**: Difficulty: medium, Validity: invalid, Tools: 4
**Tools-required**: [Order Status Fetcher, Order Performance Calculator, Execution Quality Analyzer, Regulatory Report Generator]

**Task 9**: Full execution analysis for ORD_001: status, fills, latency, VWAP benchmark, quality, position reconcile, dashboard.
**Info**: Difficulty: complex, Validity: valid, Tools: 7
**Tools-required**: [Order Status Fetcher, Fill Notification Parser, Order Latency Analyzer, Order Performance Calculator, Execution Quality Analyzer, Position Reconciler, Real-time Dashboard Generator]

**Task 10**: Multi-broker risk workflow: aggregate, fetch statuses, monitor exposure, reconcile, alert, log events, update history.
**Info**: Difficulty: complex, Validity: valid, Tools: 7
**Tools-required**: [Multi-Broker Status Aggregator, Order Status Fetcher, Risk Exposure Monitor, Position Reconciler, Alert Rule Engine, Execution Event Logger, Order History Tracker]

**Task 11**: Execution analysis + auto order optimization (invalid).
**Info**: Difficulty: complex, Validity: invalid, Tools: 8
**Tools-required**: [Order Status Fetcher, Fill Notification Parser, Order Latency Analyzer, Order Performance Calculator, Execution Quality Analyzer, Position Reconciler, Real-time Dashboard Generator, Order Optimizer]

**Task 12**: Risk management + auto hedging (invalid).
**Info**: Difficulty: complex, Validity: invalid, Tools: 8
**Tools-required**: [Multi-Broker Status Aggregator, Order Status Fetcher, Risk Exposure Monitor, Position Reconciler, Alert Rule Engine, Execution Event Logger, Order History Tracker, Position Hedger]

## A.3 EXAMPLE 3 (FIELD: HEALTHCARE AND MEDICAL)

**Multi Tool Workflows**

**Field:** Healthcare and Medical → **Subfield:** Electronic Health Records Management → **Task:** Patient Registration and Demographic Data Management

**List All Tool Names and Dependencies Generated under this task**

1. Patient Identity Validator – personal identifiers → validated identity data

2. Insurance Verification Tool – insurance details → coverage verification

3. Duplicate Patient Checker – demographic data → potential duplicate matches

4. Address Standardizer – raw address → standardized address

5. Emergency Contact Validator – contact info → validated emergency contacts

6. Patient Record Creator – validated data → new patient record

7. Demographic Data Updater – patient ID + new data → updated record

8. Medical History Importer – external records → structured medical history

9. Consent Manager – consent preferences → consent documentation

10. Patient Search Engine – search criteria → matching patient records

11. Data Quality Auditor – patient records → quality assessment

12. Registration Status Tracker – registration steps → completion status

**Simple Multi-Tool Workflows:**
1. *Basic registration:* Patient Identity Validator → Address Standardizer → Patient Record Creator
2. *Quick search:* Patient Search Engine → Demographic Data Updater

**Medium Multi-Tool Workflows:**
1. *Complete new registration:* Patient Identity Validator → Address Standardizer → Insurance Verification Tool → Duplicate Patient Checker → Emergency Contact Validator → Consent Manager → Patient Record Creator → Registration Status Tracker
2. *Record update:* Patient Search Engine → Data Quality Auditor → Demographic Data Updater → Registration Status Tracker

**Complex Multi-Tool Workflows:**
1. *Full registration with history:* Patient Identity Validator → Address Standardizer → Insurance Verification Tool → Duplicate Patient Checker → Emergency Contact Validator → Medical History Importer → Consent Manager → Patient Record Creator → Data Quality Auditor → Registration Status Tracker
2. *Comprehensive data migration:* Patient Search Engine → Duplicate Patient Checker → Medical History Importer → Demographic Data Updater → Data Quality Auditor → Consent Manager

## Meta Data

**Tool Sequences**:

Simple: [ [Patient Identity Validator, Address Standardizer, Patient Record Creator], [Patient Search Engine, Demographic Data Updater] ]

Medium: [ [Patient Identity Validator, Address Standardizer, Insurance Verification Tool, Duplicate Patient Checker, Emergency Contact Validator, Consent Manager, Patient Record Creator, Registration Status Tracker], [Patient Search Engine, Data Quality Auditor, Demographic Data Updater, Registration Status Tracker] ]

Complex: [ [Patient Identity Validator, Address Standardizer, Insurance Verification Tool, Duplicate Patient Checker, Emergency Contact Validator, Medical History Importer, Consent Manager, Patient Record Creator, Data Quality Auditor, Registration Status Tracker], [Patient Search Engine, Duplicate Patient Checker, Medical History Importer, Demographic Data Updater, Data Quality Auditor, Consent Manager] ]

**Patients**:

PAT001 – John Smith, DOB: 1985-03-15, Male, Address: 123 Main St, Springfield, IL 62701, Phone: 555-123-4567, Email: john.smith@email.com, SSN: 123-45-6789, Married, Lang: English, Status: Complete

PAT002 – Maria Garcia, DOB: 1992-07-22, Female, Address: 456 Oak Ave, Chicago, IL 60601, Phone: 555-234-5678, Email: maria.garcia@email.com, SSN: 234-56-7890, Single, Lang: Spanish, Status: Incomplete

PAT003 – David Johnson, DOB: 1978-11-08, Male, Address: 789 Pine Rd, Milwaukee, WI 53202, Phone: 555-345-6789, Email: d.johnson@email.com, SSN: 345-67-8901, Divorced, Lang: English, Status: Pending

PAT004 – Sarah Williams, DOB: 1990-12-03, Female, Address: 321 Elm St, Detroit, MI 48201, Phone: 555-456-7890, Email: sarah.w@email.com, SSN: 456-78-9012, Married, Lang: English, Status: Complete

PAT005 – Robert Brown, DOB: 1965-05-17, Male, Address: 654 Cedar Ln, Columbus, OH 43215, Phone: 555-567-8901, Email: rob.brown@email.com, SSN: 567-89-0123, Widowed, Lang: English, Status: Complete

**Insurance Providers**:

Blue Cross Blue Shield – Policies: [BC123456789, BC987654321, BC456789012], Coverage: [Individual, Family, Group], Copays: [25, 35, 15], Deductibles: [1000, 1500, 500]

Aetna – Policies: [AET123456, AET789012, AET345678], Coverage: [PPO, HMO, EPO], Copays: [30, 20, 25], Deductibles: [1200, 800, 1000]

United Healthcare – Policies: [UH987654, UH123789, UH456123], Coverage: [Select, Choice, Navigate], Copays: [35, 25, 30], Deductibles: [1500, 1000, 1200]

**Emergency Contacts**:

PAT001 – Jane Smith (Spouse), Phones: [555-123-9876, 555-123-5432], Email: jane.smith@email.com

PAT002 – Carlos Garcia (Brother), Phone: 555-234-9876, Email: carlos.garcia@email.com

PAT003 – Emily Johnson (Sister), Phones: [555-345-9876, 555-345-5432], Email: emily.johnson@email.com

**Addresses**:

123 Main Street → 123 Main St, Springfield, IL 62701, Lat: 39.7817, Long: -89.6501, Status: valid

456 Oak Avenue → 456 Oak Ave, Chicago, IL 60601, Lat: 41.8781, Long: -87.6298, Status: valid

789 Pine Road → 789 Pine Rd, Milwaukee, WI 53202, Lat: 43.0389, Long: -87.9065, Status: valid

**Medical History**:

PAT001 – Source: previous_provider, Conditions: [Hypertension, Type 2 Diabetes], Medications: [Metformin, Lisinopril], Allergies: [Penicillin], Last Updated: 2024-01-15

PAT003 – Source: patient_reported, Conditions: [Asthma, Seasonal Allergies], Medications: [Albuterol Inhaler], Allergies: [Peanuts, Shellfish], Last Updated: 2024-02-10

**Consent Records**:

PAT001 – Type: data_sharing, Status: true, Scope: [treatment, payment, operations], Exp: 2025-03-15

PAT002 – Type: research_participation, Status: false

## Tasks

**Task 1**: Register Michael Thompson (DOB 1988-04-12, Address 999 Sunset Blvd, LA 90210) and create record.
**Info**: Difficulty: simple, Validity: valid, Tools: 3
**Tools-required**: [Patient Identity Validator, Address Standardizer, Patient Record Creator]

**Task 2**: Find patient Garcia, update phone to 555-999-8888.
**Info**: Difficulty: simple, Validity: valid, Tools: 2
**Tools-required**: [Patient Search Engine, Demographic Data Updater]

**Task 3**: Register new patient + schedule appointment (invalid).
**Info**: Difficulty: simple, Validity: invalid, Tools: 4
**Tools-required**: [Patient Identity Validator, Address Standardizer, Patient Record Creator, Appointment Scheduler]

**Task 4**: Create record + insurance pre-auth (invalid).
**Info**: Difficulty: simple, Validity: invalid, Tools: 3
**Tools-required**: [Patient Record Creator, Insurance Verification Tool, Pre-authorization Request Tool]

**Task 5**: Full registration for Jennifer Lopez (DOB 1975-09-25, insurance BC123456789, spouse Carlos Lopez).
**Info**: Difficulty: medium, Validity: valid, Tools: 8
**Tools-required**: [Patient Identity Validator, Address Standardizer, Insurance Verification Tool, Duplicate Patient Checker, Emergency Contact Validator, Consent Manager, Patient Record Creator, Registration Status Tracker]

**Task 6**: Search Robert Brown, audit data, update marital status to remarried, track.
**Info**: Difficulty: medium, Validity: valid, Tools: 4
**Tools-required**: [Patient Search Engine, Data Quality Auditor, Demographic Data Updater, Registration Status Tracker]

**Task 7**: Register + insurance + duplicates + appointment + email + billing (invalid).
**Info**: Difficulty: medium, Validity: invalid, Tools: 7
**Tools-required**: [Patient Identity Validator, Insurance Verification Tool, Duplicate Patient Checker, Appointment Scheduler, Email Service, Billing Generator, Patient Record Creator]

**Task 8**: Registration with pharmacy + lab integration (invalid).
**Info**: Difficulty: medium, Validity: invalid, Tools: 6
**Tools-required**: [Patient Record Creator, Insurance Verification Tool, Pharmacy Integration Tool, Lab Integration Tool, Consent Manager, Registration Status Tracker]

**Task 9**: Comprehensive registration for Thomas Anderson (DOB 1980-06-15, Aetna AET123456, brother Neo Anderson).
**Info**: Difficulty: complex, Validity: valid, Tools: 10
**Tools-required**: [Patient Identity Validator, Address Standardizer, Insurance Verification Tool, Duplicate Patient Checker, Emergency Contact Validator, Medical History Importer, Consent Manager, Patient Record Creator, Data Quality Auditor, Registration Status Tracker]

**Task 10**: Data migration for Sarah Williams (import history, update insurance, audit, update consent).
**Info**: Difficulty: complex, Validity: valid, Tools: 6
**Tools-required**: [Patient Search Engine, Duplicate Patient Checker, Medical History Importer, Demographic Data Updater, Data Quality Auditor, Consent Manager]

**Task 11**: Registration + state HIE + CDC + insurance pre-auth + scheduling + portal (invalid).
**Info**: Difficulty: complex, Validity: invalid, Tools: 8
**Tools-required**: [Patient Identity Validator, State HIE Integration, CDC Database Connector, Insurance Pre-auth Tool, Multi-specialty Scheduler, Patient Portal Creator, Patient Record Creator, Registration Status Tracker]

**Task 12**: Registration + CDS + drug interactions + alerts (invalid).
**Info**: Difficulty: complex, Validity: invalid, Tools: 7
**Tools-required**: [Patient Record Creator, Clinical Decision Support, Drug Interaction Checker, Clinical Alerts Manager, Medical History Importer, Data Quality Auditor, Registration Status Tracker]

## A.4 EXAMPLE 4 (FIELD: ECOMMERCE AND RETAIL)

**Multi Tool Workflows**

**Field:** Ecommerce and Retail → **Subfield:** Order Processing and Fulfillment → **Task:** Returns and Refunds Processing

**List All Tool Names and Dependencies Generated under this task**

1. Return Request Validator – return request data, order history → validation status, eligibility rules
2. Return Label Generator – validated return request, shipping preferences → shipping labels, tracking info
3. Return Item Inspector – returned items data, inspection criteria → item condition assessment
4. Refund Calculator – return details, pricing data, fees → refund amounts, breakdowns
5. Payment Processor – refund amounts, payment methods → payment status, transaction IDs
6. Inventory Updater – returned items, condition assessments → inventory adjustments
7. Return Status Tracker – return IDs, status updates → current status, history
8. Customer Notifier – return status, customer info → notification confirmations
9. Return Analytics Reporter – return data, time periods → analytics reports, trends
10. Restocking Assessor – item conditions, restocking criteria → restocking decisions
11. Exception Handler – problematic returns, escalation rules → resolution recommendations
12. Return Policy Checker – product info, purchase dates → policy compliance, restrictions

**Simple Multi-Tool Workflows:**
1. *Basic return validation:* Return Policy Checker → Return Request Validator → Customer Notifier

2. *Simple refund processing:* Refund Calculator → Payment Processor → Customer Notifier

**Medium Multi-Tool Workflows:**

1. *Standard return flow:* Return Request Validator → Return Label Generator → Return Item Inspector → Refund Calculator → Payment Processor → Inventory Updater → Customer Notifier

2. *Return analytics workflow:* Return Status Tracker → Return Analytics Reporter → Exception Handler

**Complex Multi-Tool Workflows:**

1. *Complete return processing:* Return Policy Checker → Return Request Validator → Return Label Generator → Return Status Tracker → Return Item Inspector → Restocking Assessor → Refund Calculator → Payment Processor → Inventory Updater → Customer Notifier → Return Analytics Reporter

2. *Exception handling workflow:* Return Request Validator → Return Item Inspector → Exception Handler → Restocking Assessor → Refund Calculator → Payment Processor → Customer Notifier → Return Status Tracker

## Meta Data

**Customers**:

1. CUST001 – John Smith, john.smith@email.com, +1-555-0101, 123 Main St, New York, NY 10001, Segment: premium, Prefs: [email, sms]

2. CUST002 – Sarah Johnson, sarah.j@email.com, +1-555-0102, 456 Oak Ave, Los Angeles, CA 90210, Segment: regular, Prefs: [email]

3. CUST003 – Mike Davis, mike.davis@email.com, +1-555-0103, 789 Pine Rd, Chicago, IL 60601, Segment: new, Prefs: [push_notification]

**Orders**:

ORD001 – CUST001, 2023-11-01, $299.99, tax: 8%, ship: 9.99, credit_card (TXN001), Items: [Wireless Headphones x1, Phone Case x2]

ORD002 – CUST002, 2023-10-15, $89.99, tax: 7%, ship: 5.99, paypal (TXN002), Items: [Bluetooth Speaker x1]

ORD003 – CUST003, 2023-12-01, $199.99, tax: 9%, ship: 0.00, debit_card (TXN003), Items: [Smart Watch x1]

**Return Policies**:

Electronics – 30 days, 15% fee, [original_packaging, all_accessories]

Accessories – 60 days, 0% fee, [sellable_condition]

Wearables – 14 days, 10% fee, [original_packaging, no_damage]

**Warehouses**:

WH001 – New York Distribution Center, 500 Industrial Blvd, Queens, NY 11101

WH002 – California Fulfillment Center, 1000 Logistics Way, Long Beach, CA 90802

**Shipping Carriers**:

UPS – [standard: 7.99, expedited: 15.99, overnight: 29.99]

FedEx – [standard: 8.99, expedited: 16.99, overnight: 34.99]

USPS – [standard: 5.99]

**Return Requests**:

RET001 – ORD001, CUST001, 2023-11-15, defective, [ITEM001], status: initiated

RET002 – ORD002, CUST002, 2023-11-10, changed_mind, [ITEM003], status: approved

**Inspectors**:

INS001 – Quality Inspector A, WH001, auth: [basic, detailed, quality_assurance]

INS002 – Quality Inspector B, WH002, auth: [basic, detailed]

**Inventory**:

ITEM001 – stock: 50, WH001, new, cost: 75.00

ITEM002 – stock: 100, WH001, new, cost: 15.00
ITEM003 – stock: 25, WH002, new, cost: 40.00

## Tasks

**Task 1**: Validate a return request for CUST001 (ORD001 headphones defective), notify customer.
**Info**: Difficulty: easy, Validity: valid, Tools: 3
**Tools-required**: [Return Policy Checker, Return Request Validator, Customer Notifier]

**Task 2**: Calculate refund for ORD002 Bluetooth Speaker, process payment, notify CUST002.
**Info**: Difficulty: easy, Validity: valid, Tools: 3
**Tools-required**: [Refund Calculator, Payment Processor, Customer Notifier]

**Task 3**: Generate return analytics and reorder inventory (invalid).
**Info**: Difficulty: easy, Validity: invalid, Tools: 5
**Tools-required**: [Return Analytics Reporter, Inventory Reorder Tool, Trend Analyzer, Auto Purchase Tool, Supplier Notifier]

**Task 4**: Handle international return + customs docs (invalid).
**Info**: Difficulty: easy, Validity: invalid, Tools: 4
**Tools-required**: [International Return Handler, Customs Documentation Tool, Currency Converter, International Payment Processor]

**Task 5**: Full return lifecycle for ORD001 headphones.
**Info**: Difficulty: medium, Validity: valid, Tools: 7
**Tools-required**: [Return Request Validator, Return Label Generator, Return Item Inspector, Refund Calculator, Payment Processor, Inventory Updater, Customer Notifier]

**Task 6**: Track RET002 status, generate analytics, handle exceptions.
**Info**: Difficulty: medium, Validity: valid, Tools: 3
**Tools-required**: [Return Status Tracker, Return Analytics Reporter, Exception Handler]

**Task 7**: Bulk returns for 50 customers with auto-approve (invalid).
**Info**: Difficulty: medium, Validity: invalid, Tools: 8
**Tools-required**: [Bulk Return Processor, Auto Approval Engine, Bulk Refund Calculator, Store Credit Issuer, Mass Inventory Updater, Bulk Customer Notifier, Policy Override Tool, Automatic Validator]

**Task 8**: Integrate return data with CRM and dashboards (invalid).
**Info**: Difficulty: medium, Validity: invalid, Tools: 6
**Tools-required**: [CRM Integration Tool, Multi-Platform Sync, Customer Profile Merger, Executive Dashboard Generator, Data Warehouse Connector, Business Intelligence Tool]

**Task 9**: Full return lifecycle for ORD003 Smart Watch (FedEx expedited).
**Info**: Difficulty: complex, Validity: valid, Tools: 11
**Tools-required**: [Return Policy Checker, Return Request Validator, Return Label Generator, Return Status Tracker, Return Item Inspector, Restocking Assessor, Refund Calculator, Payment Processor, Inventory Updater, Customer Notifier, Return Analytics Reporter]

**Task 10**: Handle problematic return for RET001 headphones.
**Info**: Difficulty: complex, Validity: valid, Tools: 8
**Tools-required**: [Return Request Validator, Return Item Inspector, Exception Handler, Restocking Assessor, Refund Calculator, Payment Processor, Customer Notifier, Return Status Tracker]

**Task 11**: Predict return likelihood + dynamic pricing (invalid).
**Info**: Difficulty: complex, **Validity: invalid**, Tools: 15
**Tools-required**: [ML Prediction Engine, Dynamic Pricing Tool, Manufacturing Integration, Quality Improvement Analyzer, Predictive Analytics Platform, Executive Dashboard Creator, Data Mining Tool, Pattern Recognition System, Automated Decision Engine, Cross-Platform Integrator, Real-time Analytics Processor, Business Intelligence Suite, Advanced Reporting Engine, Strategic Planning Tool, Performance Optimization System]

**Task 12**: Blockchain + crypto refunds + AR/VR inspections (invalid).
**Info**: Difficulty: complex, **Validity: invalid**, Tools: 12

**Tools-required**: [Blockchain Verification System, Cryptocurrency Payment Processor, Social Media Integrator, Sentiment Analysis Engine, AR Return Inspector, VR Inspection Platform, Digital Asset Manager, Distributed Ledger Tool, Social Listening Platform, Virtual Reality Processor, Augmented Reality Engine, Digital Currency Exchange]

# B    TOOL EXAMPLES

## B.1    EXAMPLE 1 (FIELD: CUSTOMER SUPPORT)

**Field to Tool Evolution**

**Field:** Customer Support → **Sub Domain**: Ticket Management → **Task:** Track and update ticket status throughout resolution lifecycle → **Tool:** Ticket Status Updater

**The Generated Tool**

**Tool Name:** Ticket Status Updater
**Description:** Updates the status of a support ticket and validates the status transition according to business rules.
**Parameters:**

- **ticket_id** (string, required) – Unique identifier for the support ticket
- **new_status** (string, required) – Target status: `open`, `in_progress`, `pending_customer`, `pending_internal`, `resolved`, `closed`, `cancelled`
- **agent_id** (string, required) – ID of the agent making the status change
- **update_reason** (string, optional, default = `null`) – Optional reason for the status change

**Error Messages:**

- Invalid ticket ID: The provided `ticket_id` does not exist in the system. Verify the ticket number and try again.
- Invalid status transition: The requested status change is not allowed from the current status. Check valid transitions in the workflow rules.
- Agent authorization failed: The specified `agent_id` does not have permission to update this ticket. Ensure the agent has proper access rights.
- Ticket is locked: The ticket is currently being modified by another user. Wait a moment and retry the operation.

**Usage:** Provide `ticket_id`, `new_status`, and `agent_id` to update ticket status. Optionally include `update_reason` for audit purposes. The tool validates the transition before applying changes.
**Output Details:**

- **success** (boolean) – Indicates if the status update was successful
- **previous_status** (string) – The status before the update
- **current_status** (string) – The new status after the update
- **timestamp** (string) – When the status change occurred
- **updated_by** (string) – Agent ID who made the change

## B.2    EXAMPLE 2 (FIELD: ECOMMERCE AND RETAIL)

**Field to Tool Evolution**

**Field:** Ecommerce and Retail → **Sub Domain**: Product Catalog Management → **Task:** Product Quality Assurance and Content Validation → **Tool:** Image Quality Analyzer

**The Generated Tool**

**Tool Name:** Image Quality Analyzer
**Description:** Analyzes product images for quality metrics including resolution, composition, lighting, background, and technical specifications to ensure catalog standards.

**Parameters:**

- **image_urls** (array of strings, required, 1–20 items) – URLs or file paths of images to analyze
- **min_resolution** (integer, optional, default = 800) – Minimum required resolution in pixels
- **check_background** (boolean, optional, default = `true`) – Whether to analyze background cleanliness

**Error Messages:**

- Image not accessible: One or more image URLs could not be accessed or loaded.
- Unsupported image format: Images must be in JPEG, PNG, or WebP format.
- Image resolution too low: Image resolution is below the specified minimum requirement.
- Image processing failed: Technical error occurred while analyzing image quality.

**Usage:** Provide an array of `image_urls` to analyze. Optionally set `min_resolution` and background checking preferences. Returns comprehensive quality analysis for each image.

**Output Details:**

- **overall_score** (number) – Overall image quality score from 0 to 100
- **image_analyses** (array of strings) – Individual analysis results for each image
- **quality_issues** (array of strings) – Identified quality problems across all images
- **recommendations** (array of strings) – Suggestions for improving image quality

## B.3 EXAMPLE 3 (FIELD: FINANCIAL TRADING)

**Field to Tool Evolution**

**Field:** Financial Trading → **Sub Domain**: Risk Management and Assessment → **Task:** Portfolio risk analysis and stress testing → **Tool:** Correlation Matrix Calculator

**The Generated Tool**

**Tool Name:** Correlation Matrix Calculator
**Description:** Calculates correlation matrices between assets using historical return data with various correlation methods and time windows.

**Parameters:**

- **calculation_method** (string, required) – Correlation calculation method: `pearson`, `spearman`, `kendall`
- **time_window** (integer, optional, default = 252) – Rolling window size in days for correlation calculation (30–1000)
- **min_periods** (integer, optional, default = 20) – Minimum number of observations required for correlation calculation

**Error Messages:**

- Invalid calculation method: Use `pearson`, `spearman`, or `kendall` correlation methods only.
- Insufficient data: Not enough price data to calculate correlations. Ensure at least `min_periods` observations per asset.
- Invalid time window: `time_window` must be between 30 and 1000 days.
- Singular matrix: Correlation matrix is singular, indicating perfect correlation between some assets.

**Usage:** Specify correlation calculation method and optionally adjust `time_window` and `min_periods`. Requires price data from Historical Price Fetcher.

**Output Details:**

- **correlation_matrix** (array of numbers) – Flattened correlation matrix values
- **asset_pairs** (array of strings) – Asset pair labels corresponding to correlation values
- **highest_correlation** (number) – Highest correlation coefficient found
- **lowest_correlation** (number) – Lowest correlation coefficient found
- **matrix_rank** (integer) – Rank of the correlation matrix

## B.4 EXAMPLE 4 (FIELD: HEALTHCARE AND MEDICAL)

**Field to Tool Evolution**

**Field:** Healthcare and Medical → **Sub Domain**: Patient Registration and Scheduling → **Task:** Appointment scheduling and calendar management across multiple providers → **Tool:** Wait List Manager

**The Generated Tool**

**Tool Name:** Wait List Manager
**Description:** Manages patient wait lists for unavailable appointment slots, automatically matching patients with newly available appointments based on preferences and priorities.

**Parameters:**

- **action** (string, required) – Action to perform: `add`, `remove`, `check_matches`, `get_list`
- **patient_name** (string, optional, default = None) – Patient name (required for `add` action)
- **patient_phone** (string, optional, default = None) – Patient phone number (required for `add` action)
- **preferred_providers** (array of strings, optional, default = None) – List of preferred provider IDs
- **preferred_dates** (array of strings, optional, default = None) – List of preferred dates in ISO 8601 format
- **wait_list_id** (string, optional, default = None) – Wait list entry ID (required for `remove` action)

**Error Messages:**

- Invalid action: Action must be one of [`add`, `remove`, `check_matches`, `get_list`]
- Missing patient info: `patient_name` and `patient_phone` are required for `add` action
- Wait list entry not found: No wait list entry exists with the specified `wait_list_id`
- Invalid provider IDs: One or more `preferred_providers` do not exist
- Invalid date format: `preferred_dates` must be in ISO 8601 format
- Duplicate entry: Patient is already on wait list for this provider/time combination

**Usage:** Specify `action` and provide required parameters. Use `add` to put patients on wait list, `check_matches` to find available appointments, and `remove` to take patients off wait list.

**Output Details:**

- **wait_list_id** (string) – Unique identifier for wait list entry
- **status** (string) – Operation status
- **matches_found** (integer) – Number of matching appointments found
- **matched_slots** (array of strings) – List of available appointment slots that match criteria

## B.5 EXAMPLE 5 (FIELD: TRANSPORTATION AND LOGISTICS)

---

**Field to Tool Evolution 5**

**Field:** Transportation and Logistics → **Sub Domain**: Freight and Cargo Management → **Task:** Route planning and optimization for freight shipments → **Tool:** Traffic Condition Analyzer

---

**The Generated Tool**

**Tool Name:** Traffic Condition Analyzer
**Description:** Analyzes current and predicted traffic conditions along route segments to estimate delays and optimal departure times.

**Parameters:**

- **route_segments** (array of strings, required, 1–50 items) – Array of route segment identifiers or coordinate pairs
- **departure_time** (string, required, ISO 8601) – Planned departure time in ISO 8601 format
- **analysis_duration_hours** (integer, optional, default = 24) – How many hours ahead to analyze traffic patterns (1–72)
- **day_of_week** (string, required) – Day of week for traffic pattern analysis: `monday`, `tuesday`, `wednesday`, `thursday`, `friday`, `saturday`, `sunday`

**Error Messages:**

- Invalid departure time: Provide valid ISO 8601 formatted date-time string.
- Invalid day of week: Use full day name in lowercase (`monday` through `sunday`).
- Too many route segments: Maximum 50 segments allowed for analysis.
- Invalid analysis duration: Duration must be between 1 and 72 hours.
- Traffic data unavailable: Unable to retrieve traffic information for specified route segments.

**Usage:** Provide `route_segments`, `departure_time`, and `day_of_week`. Optionally specify `analysis_duration_hours`. Returns traffic predictions and delay estimates for route optimization.

**Output Details:**

- **total_delay_minutes** (number) – Total expected traffic delay in minutes
- **congestion_segments** (array of strings) – Route segments expected to have heavy congestion
- **optimal_departure** (string) – Recommended departure time to minimize delays
- **traffic_severity** (string) – Overall traffic severity level: `low`, `moderate`, `high`, `severe`

---

# C PIPELINE DETAILS

## C.1 FAILURE AND SUCCESS MODELS OF TOOL CALLS

---

**Failure mode 1: Schema mismatch**

**Tool call message:**
*Insurance_Information_Updater* (patient_id = 'PAT001', insurance_fields = [], insurance_values = ['Blue Cross'])

**Response:**
*Status*: FAIL, *Status Code*: 400, *Error Message*: Invalid parameter: insurance_fields array must contain at least 1 item."

---

**Failure mode 2: Parameter inconsistency**

**Tool call message:**
*Insurance_Information_Updater* (patient_id = 'PAT001', insurance_fields = ['provider', 'policy_number'], insurance_values = ['Blue Cross'])

**Response:**
*Status*: FAIL, *Status Code*: 400, *Error Message*: Mismatched fields and values: Ensure insurance_fields and insurance_values arrays have the same length.

---

| Failure mode 3: Response inconsistency | Success mode: Correct response |
|---|---|
| **Tool call message:**
*Regulation_Detail_Fetcher* (regulation_id = 'INVALID-REG-999')

**Response:**
*Status*: PASS, *Status Code*: 200, *Return Data*: Error: Invalid regulation ID: Ensure the regulation ID is correct and exists in the database. | **Tool call message:**
*Insurance_Information_Updater* (patient_id = 'PAT001', insurance_fields = ['provider'], insurance_values = ['Blue Cross'])

**Response:**
*Status*: PASS, *Status Code*: 200, *Return Data*: update_status: Success, updated_insurance: ['provider'] |

### C.2 PROMPTS

We post our prompts under this anonymized repository: `https://anonymous.4open.science/r/SynthTools-44C6/README.md`

## D DEDUPLICATION DETAILS

To eliminate near-duplicate tools generated through hierarchical evolution, we employ a multi-stage de-duplication pipeline. The process begins with exact de-duplication applied independently within each domain/field. We first normalize and compare the `tool_name` attributes, followed by the `tool_body`, which encompasses the description, parameters, usage, and output schema. This step reduces redundancy and prevents unnecessary computation in subsequent stages.

We then implement a semantic de-duplication pipeline, adapted from SemDepDup—a state-of-the-art method proposed by Abbas et al. (2023)—tailored to the structural characteristics of our dataset. The semantic phase proceeds as follows:

We construct an embedding-based similarity graph over the normalized `tool_name` and `tool_body` fields. Specifically, let $e(t) \in \mathbb{R}^d$ denote the embedding of tool $t$, and define the coine similarity between tool $t_i$ and $t_j$ as $S_{ij} = \cos\big(e(t_i), e(t_j)\big)$. An adjancency matrix is then defined as $A_{ij} = \mathbf{1}\{S_{ij} \geq \tau\}$, $\tau \in (0, 1)$, and the corresponding undirected similarity graph is $\mathcal{G} = \big\{(i, j) : A_{ij} = 1\big\}$. Connected components in $\mathcal{G}$ represent candidate duplicate sets: singleton nodes are considered unique tools, whereas multi-node components are treated as clusters of near-duplicates. We then apply a selection algorithm (Algorithm 1) to retain representative tools from each component.

Empirically, cross-field duplication is rare, as workflows and vocabularies tend to differ significantly across domains (Figure 8). Most residual redundancy occurs between adjacent tasks and subdomains, where tool functionalities partially overlap. The degree of de-duplication depends on the threshold $\tau$; Figure 9 illustrates the elimination rate as a function of $\tau$. Based on a validation sweep that balances compactness and coverage, we set $\tau = 0.85$ in practice.

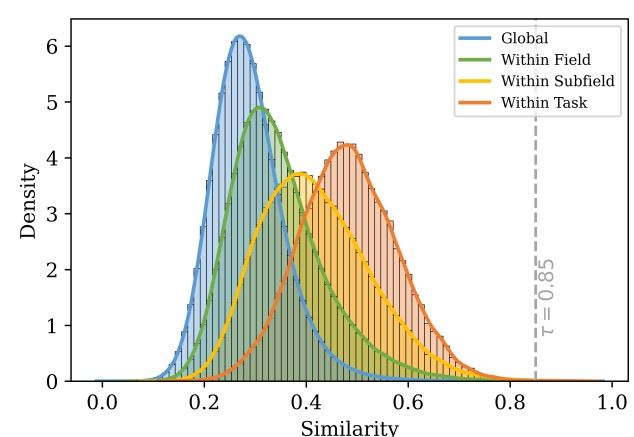

Figure 8: Distribution of semantic similarity scores between tools at varying levels of domain granularity: across fields (blue), within a field (green), within a subfield (yellow), and within a task (red).

---

**Algorithm 1** Selection rule for connected components

---

1: **for** each connected component $C \subseteq \{1, \ldots, |T|\}$ **do**
2:     **if** $|C| = 1$ **then**
3:         retain the tool unchanged.
4:     **else if** $|C| = 2$ **then**
5:         choose one uniformly at random (with a fixed PRNG seed for reproducibility), discard the other.
6:     **else**                                                      $\triangleright\, |C| \geq 3$
7:         Initialize $L \leftarrow C$.
8:         **while** $\exists\, i \neq j \in L$ with $u_i^\top u_j \geq \tau$ **do**
9:             **for** each $i \in L$ **do**
10:                 Compute degree

$$\deg_\tau(i) = \sum_{j \in L \setminus \{i\}} A_{ij}$$

11:                 Compute incident-sum

$$w_\tau(i) = \sum_{j \in L \setminus \{i\}} A_{ij}\, (u_i^\top u_j)$$

12:             **end for**
13:             Select node to drop by lexicographic maximization:

$$v^\star \in \arg\max_{i \in L} \big( \deg_\tau(i),\, w_\tau(i) \big)$$

14:             Let

$$k^\star = \arg\max_{j \in L \setminus \{v^\star\}} u_{v^\star}^\top u_j$$

15:             Check condition $u_{v^\star}^\top u_{k^\star} \geq \tau$.
16:             Drop $v^\star$: update $L \leftarrow L \setminus \{v^\star\}$.
17:         **end while**
18:         Return survivors $L$ for component $C$.
19:     **end if**
20: **end for**

---

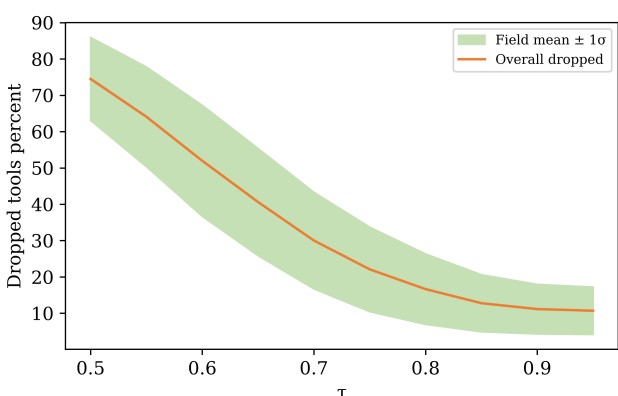

Figure 9: Percentage of tools dropped v/s similarity score threshold

# E   DATASET DETAILS

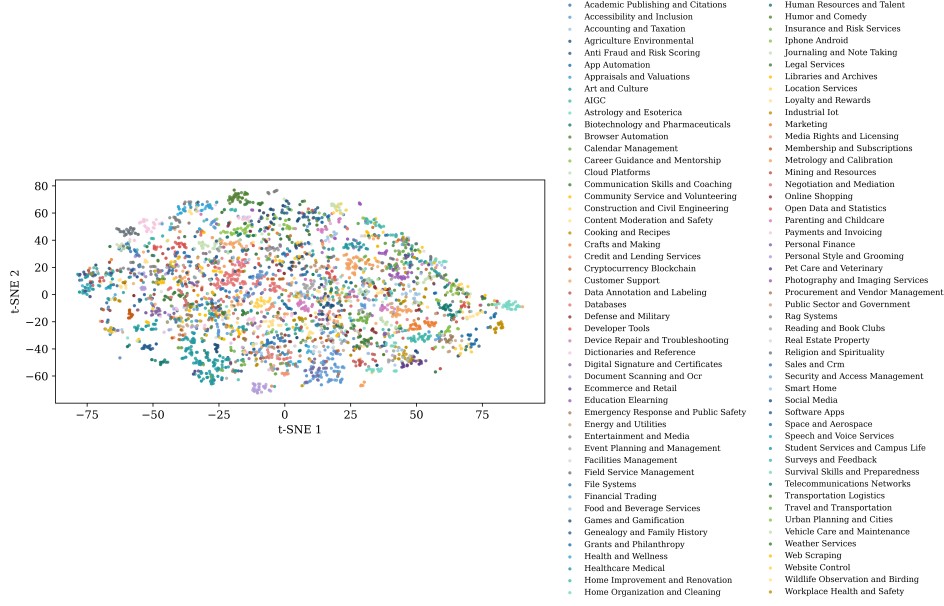

Figure 10: Distribution of tool embeddings across 100 fields

# F   LLM USE

LLMs were used in both polishing the writing and in search for related works. LLMs were also used in refining the prompts for the LLM studied in this work. We use Claude-Sonnet-4 for our experiments and to refine our writing

