# OpenReview forum: "SynthTools: A Framework for Scaling Synthetic Tools for Agent Development"
_ICLR.cc/2026/Conference — ICLR 2026 Conference Withdrawn Submission_

### Official Review · Reviewer_Wbgh · 2025-10-23

**Soundness:** 2
**Presentation:** 2
**Contribution:** 2
**Rating:** 4
**Confidence:** 3

**Summary:**

The SynthTools, a scalable framework for generating synthetic tool ecosystems to address the bottleneck: the scarcity, instability, and practical limitations of real-world APIs. It provides a methods to generate stable, controllable, and domain-agnostic environment. The framework consists of three core components: (1) Tool Generation (2) Tool Simulation, with parameter validation and response generation, and (3) Tool Audit, a quality control pipeline with LLM-as-a-judge to verify simulator correctness against automatically generated test cases.

The framework demonstrates massive scalability, generating thousands of tools across over 100 domains. This framework builds an automated audit closed loop of "generating test cases → simulating responses → evaluating results". It verifies the reliability of the entire quality control system through the judges themselves, providing a confidence guarantee for the large-scale generation of high-quality tools

**Strengths:**

1. Scalability and diversity: it outperforms existing benchmarks in both the number of domains and tools per domain, enabling training on a scalable synthetic data previously unachievable.
2. Reliability and Stability: by a robust, multi-stage quality control process, featuring a tool simulator with 93.6% accuracy and an LLM judge validated at 99% accuracy with a 0% false-positive rate over 300 stress test cases, which is critical for trustworthy evaluation.
3. Hierarchical generation method: ensures that the synthetic tools are not random but are grounded in meaningful control workflows. Also deduplication pipeline provide more details on the systematically way to retain representative tools from each component.

**Weaknesses:**

1. There is a trade-off: the framework achieves scalability by abstracting away implementation-level fidelity, potentially leaving LLM unprepared for real-world execution challenges, such as network unstable, API update or other errors. Want to see more discussion on the gap between synthetic data and real-world API.
2. As mentioned in paper, the systemic risk of its LLM-as-judgle cannot be ignore. The stress test case not eliminate my concern on the final quality of synthetic data quality. Also, I am little confused why use stress test case here, rather than other kind of test. The paper didn't explain the backend program setting for stress test, which makes stress test make non-sense for final conclusion. Any inherent bias or conceptual blind spot in the base model could be propagated and reinforced throughout the pipeline.
3. Some model training experiments on data and the model performance improvement will be helpful to prove the effectiveness of the method and the quality of the data. More cost details analysis of the synthetic pipeline also can enrich the article.

**Questions:**

Please check weaknesses.

---

### Official Review · Reviewer_gx5t · 2025-10-30

**Soundness:** 3
**Presentation:** 3
**Contribution:** 2
**Rating:** 2
**Confidence:** 4

**Summary:**

The paper presents SynthTools, a framework for using LLMs to generate synthetic tools and simulate their use. SynthTools has 3 stages: tool generation, tool simulation, and tool auditing. The authors present the prompting methods for each and validate the results are aligned with objectives of diversity and correctness.

**Strengths:**

## Well-motivated, simple design
SynthTools follows a reasonable method of using LLMs to generate ideas at scale, validate those ideas align with some values of diversity, and simulate the tools. Using LLMs to simulate tools rather than relying on real environments or even engineered simulations is well-motivated.

## Evaluation of each SynthTools component
The authors designed appropriate evaluation for their system, and the results provide assurance in the quality of SynthTools.
In the simulation component of SynthToools, the resulting prompt framework + LLM is shown to align well with an existing tool use benchmark (ACEbench). In the audit component, the prompt + LLM passes stress-tests manually curated by the authors.

**Weaknesses:**

## Missing demonstrated value
While the motivation for SynthTools is clear, as a method for making environments with various tools and constraints to train AI systems in long-horizon tool-using tasks, these results are missing from the paper itself. Evaluation is on the correctness of subcomponents of the system, but there is no evaluation on training an actual AI system in the ecosystems generated by SynthTools. Even within the existing evaluation of SynthTools components, there seems to be no baselines to compare to-- for example: showing that the diversity using the hierarchical domain decomposition is better than that of prompting with CoT or other well-established simple methods. Or showing how these evaluation numbers change when using the SynthTools framework across different backbone LLMs. Without evaluations showing the value of actually using SynthTools, especially in comparison to existing alternatives, it is difficult for me to advocate for the value of the paper as-is.

## Quality of generated tools?
How do we know the synthesized tools are useful? Figure 6 shows a t-SNE graph suggesting that generated tools cover different embedding spaces, but this does not necessarily suggest that those tools are anything useful for the tasks at hand, and perhaps more importantly, for training AI systems to operate in realistic environments.

**Questions:**

* L115, 155, 425 say that tools are encouraged to be composable. Is the simulator also encouraged to simulate tools in this composable manner? i.e. composing tool call outputs, or calling to symbolic components to execute simple tool calls
* How did the authors verify the LLM-as-a-judge stress tests in L365? Are the test cases effectively challenging and diverse? It would be nice to include statistics demonstrating the validity of this micro-benchmark, and maybe a few examples in the appendix as well.

---

### Official Review · Reviewer_XjES · 2025-10-31

**Soundness:** 3
**Presentation:** 3
**Contribution:** 2
**Rating:** 2
**Confidence:** 3

**Summary:**

This paper proposes a framework that allows the creation of tools in a purely synthetic fashion. The authors scale this to a diverse and large (1000 tools) pool of tools that could be used in environments to either test, synthesize rollouts or use in an RL setting

**Strengths:**

They tackle a key component in modern LLM agent development, which is the creation of diverse enough tools. The way this is done is quite sound, with many internal checks and validators. The final analysis also shows that the generated set of tools is diverse (a key aspect)

**Weaknesses:**

It is not totally clear that scaling the number of tools is a right approach for LLM agents. Valid alternatives are minimizing the tools (such as only computer use, bash or web search) or creating tools on the fly as needed for one domain. This is just on the motivation side, and does not affect the quality of the work itself

The biggest question I have is "so what?". This paper shows how to create a large and diverse set of tools. What is not clear is how this is valuable in the end. One obvious experiment that could show that value is to train on environments using those tools, and using the obtained LLM to solve other established tool benchmarks. Another one is to use that resulting dataset as evaluation and showing that modern frontier LLMs cannot solve it yet

**Questions:**

my biggest question is on additional experiments which would add more value to this paper but might be too tight for the rebuttal phase

---

### Official Review · Reviewer_3TAc · 2025-11-01

**Soundness:** 2
**Presentation:** 2
**Contribution:** 2
**Rating:** 4
**Confidence:** 3

**Summary:**

This paper presents SynthTools, a framework for generating synthetic tool ecosystems to support LLM agent development. The framework comprises three components: Tool Generation (hierarchical domain evolution from fields to tools), Tool Simulation (emulating realistic tool behaviors), and Tool Audit (quality control through test case generation and LLM-based verification). The authors demonstrate scalability across 100+ fields with up to 1,000 tools per field, achieving 93% tool simulation accuracy and 99% LLM judge accuracy.

**Strengths:**

The three-component pipeline (generation, simulation, audit) is well-motivated and addresses key challenges systematically.
The validation against ACEBench (94% accuracy) and manual stress testing (99% judge accuracy) provides concrete evidence of reliability.

**Weaknesses:**

The paper's central claim is that agents trained on synthetic tools can transfer learned capabilities to real-world interfaces. While cited references (Li et al., 2023; Kimi, 2025; Sullivan et al., 2025) provide "supporting evidence," the paper itself conducts no experiments demonstrating this transfer. This is a critical gap—the framework's utility fundamentally depends on whether synthetic tool training actually improves real-world agent performance. I would suggest the authors include experiments showing agent performance on real-world benchmarks (e.g., ToolBench, API-Bank) after training on SynthTools, and comparison with agents trained on real APIs or other synthetic approaches, and analysis of which synthetic tool characteristics correlate with transfer effectiveness.

The paper would benefit from engaging with broader literature on tool learning and synthetic data generation. For example, recent work has explored unified approaches to tool retrieval and generation that could inform the framework's design (e.g., ToolGen and similar systems that jointly optimize retrieval and execution). Understanding how tools are discovered and composed in practice would strengthen the generation component.

**Questions:**

I think tool simulation evaluation is Incomplete. The 93-94% simulation accuracy is impressive, but there is no analysis of failure modes: What types of errors occur in the 6-7% of cases? Are they systematic (e.g., specific parameter types, tool categories)?

Section 6.1 mentions tasks are generated using the hierarchical procedure,  while how are task difficulties (simple/medium/complex) determined? What ensures tasks actually require the specified tool chains?

---

### Note · Authors · 2025-11-24

I have read and agree with the venue's withdrawal policy on behalf of myself and my co-authors.